# Ancient genomes reveal social and genetic structure of Late Neolithic Switzerland

Anja Furtwängler[1], A. B. Rohrlach [2,3], Thiseas C. Lamnidis[2], Luka Papac[2], Gunnar U. Neumann[1,2], Inga Siebke[4], Ella Reiter[1], Noah Steuri[5], Jürgen Hald[6], Anthony Denaire[7], Bernadette Schnitzler[8], Joachim Wahl[9,10], Marianne Ramstein [11], Verena J. Schuenemann[1,12,13], Philipp W. Stockhammer [2,14], Albert Hafner [5,15], Sandra Lösch [4], Wolfgang Haak [2], Stephan Schiffels [2] & Johannes Krause [1,2,12✉]

Genetic studies of Neolithic and Bronze Age skeletons from Europe have provided evidence for strong population genetic changes at the beginning and the end of the Neolithic period. To further understand the implications of these in Southern Central Europe, we analyze 96 ancient genomes from Switzerland, Southern Germany, and the Alsace region in France, covering the Middle/Late Neolithic to Early Bronze Age. Similar to previously described genetic changes in other parts of Europe from the early 3rd millennium BCE, we detect an arrival of ancestry related to Late Neolithic pastoralists from the Pontic-Caspian steppe in Switzerland as early as 2860–2460 calBCE. Our analyses suggest that this genetic turnover was a complex process lasting almost 1000 years and involved highly genetically structured populations in this region.

[1] Institute for Archaeological Sciences, Archaeo- and Palaeogenetics, University of Tübingen, Tübingen, Germany. [2] Max Planck Institute for the Science of Human History, Jena, Germany. [3] ARC Centre of Excellence for Mathematical and Statistical Frontiers, School of Mathematical Sciences, The University of Adelaide, Adelaide, SA 5005, Australia. [4] Department of Physical Anthropology, Institute of Forensic Medicine, University of Bern, Bern, Switzerland. [5] Institute of Archaeological Sciences, Prehistoric Archaeology, University of Bern, Bern, Switzerland. [6] Archaeological Office of the District of Constance, Konstanz, Germany. [7] Department of history of arts and Archaeology, University of Burgundy, Burgundy, France. [8] Museum of Archaeology Strasbourg, Strasbourg, France. [9] Institute for Archaeological Science, Palaeoanthropology, Eberhard Karls University Tübingen, Tübingen, Germany. [10] State Office for Cultural Heritage Management Baden-Wuerttemberg, Konstanz, Germany. [11] Archaeological Service of the canton of Bern, Bern, Switzerland. [12] Senckenberg Centre for Human Evolution and Palaeoenvironment, University of Tübingen, Tübingen, Germany. [13] Institute of Evolutionary Medicine, University of Zurich, Zurich, Switzerland. [14] Institut für Vor- und Frühgeschichtliche Archäologie und Provinzialrömische Archäologie, Ludwig Maximilian University, Munich, Germany. [15] Oeschger Centre for Climate Change Research, University of Bern, Bern, Switzerland. ✉email: krause@shh.mpg.de

Genetic studies have revealed that Central Europeans, during the Neolithic, were genetically mixed between indigenous European hunter-gatherers and new incoming people with ancestry related to Western Anatolian early farmers[1–5]. Towards the end of the Neolithic period, just before the transition to the Bronze Age, a second arrival of a new ancestry component in Europe was detected genetically[6,7], coinciding with the emergence of the Central and Eastern European Corded Ware Complex (CWC; encompassing Battle Axe and Single Grave cultural groups, ref. [8]). The new genetic component was most closely related to ancestry from the Pontic–Caspian steppe, found in individuals associated with the Yamnaya complex. While the origin of this new, third European ancestry component has been attested in many European regions[6,7,9], the exact timing of the arrival in other regions, as well as the demographic processes underlying this genetic admixture, are less clear.

Archaeologically, the Neolithic period in Switzerland is dominated by lakeshore and bog settlement sites with organic preservation, inner alpine sites of the Rhône valley, and high mountain pass sites[10]. Apart from settlement remains, stone cist graves from the Chamblandes type and a few megalithic burials towards the end of the Neolithic have been found, such as the dolmen burials of Oberbipp, Sion, Aesch, and others[10–13]. The rich archaeological record in Switzerland makes the region relevant for studies of population history in Central Europe. This is due to the particularly well-preserved wetland settlements from which the wooden parts provided one of the best dated dendrochronological records in prehistoric Europe[11].

In Switzerland, CWC finds are exclusively from settlements on the banks of the large pre-alpine lakes. They are particularly numerous in the region of Lake Zurich in Eastern Switzerland and the Three Lakes Region in Western Switzerland. The sites on Lake Neuchâtel lie on the South-Western edge of the area influenced by CWC. In Eastern Switzerland, the new, cord-ornamented ceramic style was rapidly adopted, while in Western Switzerland this process had lasted several centuries. High-precision dendrochronological data obtained from the building structures of CWC settlements provide clear approaches to absolute chronology[14].

Although there are numerous Neolithic and Early Bronze Age sites from the lakeshores and moors, there are no burials directly related to them. This is due, among other things, to the fact that the Chamblandes type stone cist tombs were already in use in the fifth millennium BCE. This burial custom, however, most probably ends around 3800 BCE, i.e., exactly at the time when the lakeshore settlements begin to become numerous. The burials of the Early Bronze Age are concentrated in inner alpine regions (Rhone valley, Lake Thun area, and foothills of the Alps) from which no lakeside settlements of this period are known. In periods with many graves, there are no settlements and vice versa, in periods with many settlements the corresponding burials are missing. The reasons for this are probably of taphonomic nature.

Thus, only four ancient genomes have been published so far for the territory of present-day Switzerland: one Late Pleistocene hunter-gatherer individual from Bichon cave[15] and three individuals associated with the Bell Beaker phenomenon from the dolmen burial of Sion-Petit-Chasseur[9].

The aim of this project is to investigate the transition from the Neolithic to the Bronze Age in Switzerland in detail, with a specific focus on the timing of the arrival, source and mixture process of steppe-related ancestry, and the social and demographic structure before and after this transition. Using relatively dense temporal sampling, we generate genome-wide data from 96 individuals dating to the Neolithic and Early Bronze Age period from Switzerland, Southern Germany, and the Alsace region in France. We also generate data from one individual from the Early Iron Age and the Roman period, respectively. We find the expected large genetic turnover at the beginning of the third millennium BC and a highly genetically structured population in the region of present day Switzerland at that time period. The predominant social structure, furthermore, was probably patrilocal.

## Results

**Ancient DNA authentication and uniparental markers**. The ancient individuals from this study originate from 13 Neolithic and Early Bronze Age sites in Switzerland (Fig. 1b), Southern Germany, and the Alsace region in France. All samples taken from the individuals were radiocarbon dated (Supplementary Note 2 and Supplementary Data 1). In a preliminary screening, 263 samples were enriched for mtDNA. We reconstructed complete mitochondrial genomes, used them to estimate DNA library contamination (Supplementary Data 1), and identified 96 samples that had less than 5% of contamination for further analyses. We determined mtDNA haplogroups using the software *haplogrep* (ref. [16], Supplementary Note 3, Supplementary Fig. 1, and Supplementary Data 1) and found the macrohaplogroups N1a, W, X, H, T2, J, U2, U3, U4, U5a, U5b, K, and U8 in our samples.

For genome-wide analysis, we genotyped all selected individuals on ~1.2 million genomic SNPs[2], also containing 49,704 SNPs on the X chromosome and 32,670 SNPs on the Y chromosome. SNPs on the X chromosome were used to estimate nuclear contamination in male individuals (Supplementary Data 1), and we again used a threshold of 5% to select clean libraries for further analysis (96 individuals). We also determined Y chromosomal haplogroups (Supplementary Note 4, Supplementary Fig. 2, Supplementary Data 1, and Supplementary Table 1).

**Population turnover at the transition to the Bronze Age**. We combined the genotype data of the new 96 individuals from this study that passed our contamination tests with 399 published ancient genomes from the same time period from Central and Western Europe as well as Neolithic individuals from Anatolia and the Pontic steppe (individuals annotated as Yamnaya Samara in ref. [6]) and genotype data of modern individuals from the POPRES[17] and the Human origins (HO)[1] datasets for various analyses.

We projected our 96 new ancient genomes from Switzerland and surrounding regions and 52 published ancient genomes selected to reflect the genetic landscape of Europe at different time points onto the first two principal components constructed from 1960 individuals of 38 European populations from the POPRES dataset (Fig. 1c and "Methods"). Two distinct clusters can be identified and were also confirmed by ADMIXTURE analysis (Supplementary Note 5), one consisting of individuals dating to 4770–2500 calBCE, and one comprising individuals dating to 2900–1750 calBCE. The oldest individuals from the sites of Niederried (CH) and Lingolsheim (F) fall close to ancient individuals from Anatolia associated with early agricultural contexts. More recent individuals from the megalithic burials at Oberbipp (CH) and Aesch (CH) are shifted further towards Western Hunter-Gatherers (WHG) and close to modern-day Sardinian individuals, as well as towards Early and Middle Neolithic individuals from Iberia or the Middle Elbe-Saale (MES) region in Central Germany. This shift mirrors an increase of hunter-gatherer-related ancestry during the middle Neolithic that has been described previously for other parts of Europe.

The second distinct cluster is shifted towards the individuals associated with the "Yamnaya" complex, similar to other European groups younger than 2700 BCE, relative to individuals

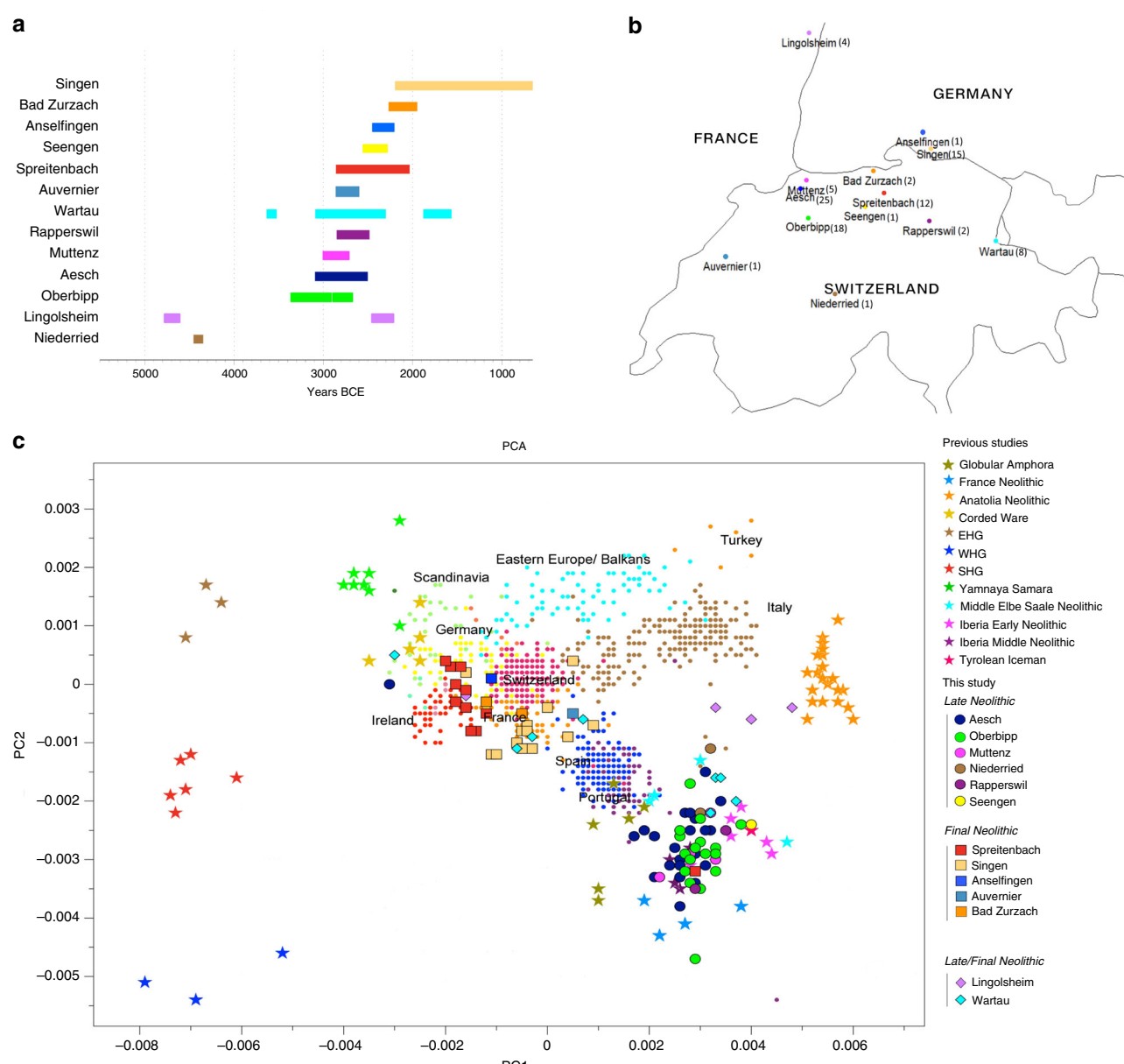

**Fig. 1 Genetic, temporal, and spatial structure of individuals in this study. a** Time ranges of calibrated radiocarbon dates of the archaeological sites.
**b** Geographical distribution of the sites and samples sizes per sites in brackets. Map generated with R version 3.4.3 (R Core Team 2017) using the CIA
World Data Bank II currently (mid 2003) available from http://www.evl.uic.edu/pape/data/WDB/. **c** PCA was reconstructed on 1960 modern European
individuals of the POPRES dataset and ancient genomes were projected onto it.

older than 2700 BCE. In this cluster, the oldest individuals are
closest to Late Neolithic groups on the steppe, whereas more
recent individuals are once again shifted towards the Middle/Late
Neolithic cluster. All Final Neolithic and Early Bronze Age
individuals fall within the range of modern-day Europeans, but
none of the newly sequenced individuals of this study overlap
with the present-day Swiss populations in this analysis, suggesting
additional population changes in the region after the Middle
Bronze Age.

Our individuals sequenced in this study fall in PC space
between WHG individuals, Western Anatolian Neolithic Farmers
(ANF) and steppe pastoralists from Samara (YAM), similar to
other Late Neolithic individuals such as the Tyrolian Iceman[18]
and Bronze Age populations such as individuals of the Bell Beaker
complex[6] from Europe. Therefore, we modeled them as a three-
way mixture between these three populations using *qpAdm* from

the ADMIXTOOLS package (ref. [19], "Methods" and Supplemen-
tary Tables 2 and Supplementary Data 4). The overall pattern
observed from this analysis matched previous analyses of that
type[9]. Individuals from older sites (Early Neolithic and Middle
Neolithic) are consistent with a two-way mixture between WHG
and ANF ancestry, whereas individuals from younger sites after
~2700 BCE exhibit substantial amounts of ancestry related to
YAM. Furthermore, the proportions of this component differ
strongly between sites and tend to decrease over time (Fig. 2a).
This trend is confirmed by further analysis of the ancestry
components on an individual level.

Compared with previous studies[2,9] analyzing Neolithic and
Bronze Age individuals from present-day Germany and Great
Britain, which do not report individuals dating to the transition
period directly, in this study we analyze a gapless time-transect
covering the Neolithic to Bronze Age transition. By viewing the

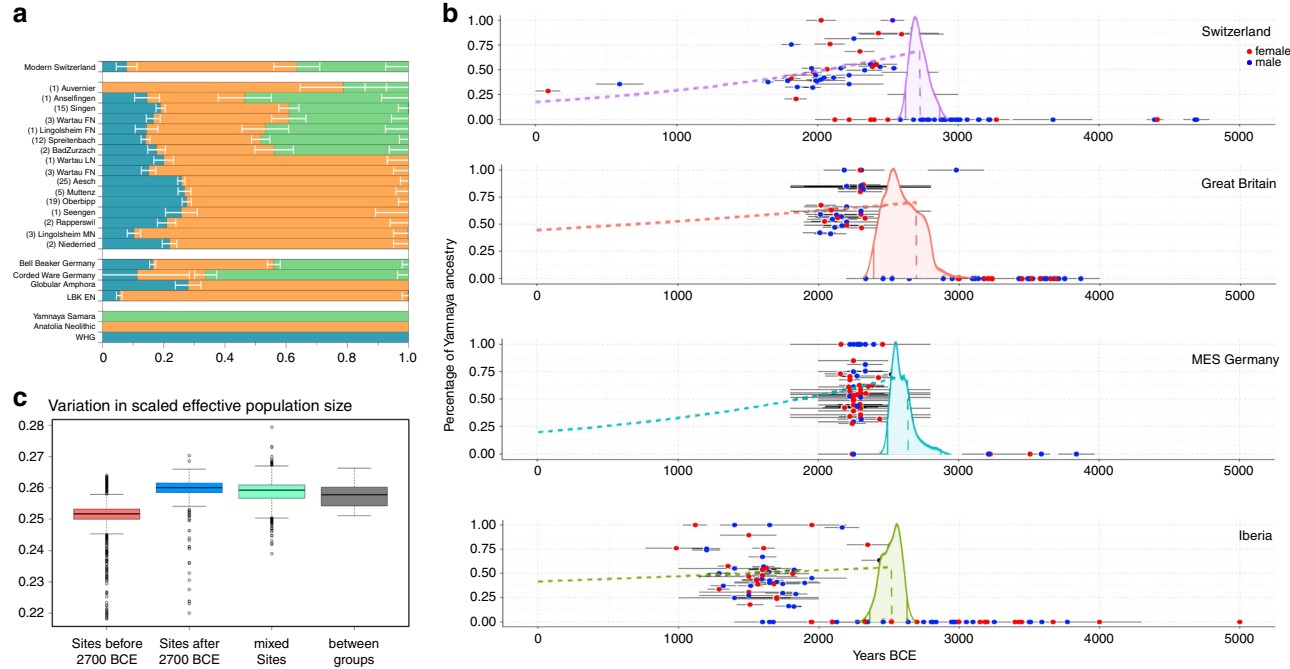

**Fig. 2 Genetic turnover at the transition to the Central European Bronze Age. a** Three-way qpAdm models of the ancient individuals from Switzerland (number of individuals in brackets) sorted by sites in chronological order (bottom to top) with the source populations WHG, steppe pastoralists (Yamnaya Samara) and Anatolia Neolithic. Error bars represent standard error of the proportion of each component. **b** Relative proportion of the steppe-related ancestry component for each individual in four different regions, calculated with qpAdm and estimates of arrival times (error bars represent the range of C[14] dating) and following decrease of the component (dashes lines). Red dots represent female individuals and blue dots male individuals. **c** An estimate of genetic diversity between individuals before 2700 BCE and after 2700 BCE and sites with individuals from both periods as well as modern European populations (German and French from the HO dataset).

YAM-related ancestry component estimated with *qpAdm* over time at an individual level, it becomes apparent that this ancestry was virtually absent before 2700 BCE, followed by a steep increase in parts of the population starting around 2700 BCE (Supplementary Note 7 and Supplementary Data 4). After this rapid increase in individual proportions of YAM-related ancestry from 0% to ~60%, a decrease down to 25–35% can be observed over the next thousand years. We also note four female individuals that can be modeled without any YAM-related ancestry even 1000 years after the appearance of that genetic component in the area. Comparing outgroup-*f3* statistics between the autosomes and the X chromosome of Final Neolithic and Bronze Age individuals we find that autosomes are more closely related to YAM-related ancestry than the X chromosomes are (Supplementary Note 6), consistent with a model in which more males than females brought YAM-related ancestry into the region as already shown by previous studies[20].

We analyzed pairwise genetic differences across all analyzed genomic positions between individuals before and after the genetic turnover and found that the mismatch rates increase, on average, by around 0.009 after 2700 BCE for all populations (Fig. 2c, see also "Methods"). This is more than twice the increase that can be attributed to the between-population variability in rates and indicates a significant increase in genetic diversity after the arrival of the YAM-related ancestry component in Central Europe. Modern populations would be expected to have higher levels on average but are not compared in this analysis since the modern individuals from published datasets usually do not originate from groups with the same background (e.g., being related distantly) as could be expected in multiple burials.

Comparing our newly analyzed individuals from Switzerland with ancient genomes from Great Britain, Iberia, and Germany[2,9,21] we modeled the arrival time of the YAM-related ancestry in the different broadly defined European regions (Fig. 2b, see "Methods"). While our models indicate that the proportions of the YAM-related ancestry peaks earlier in the Swiss dataset (around 2750 BCE) compared with the comparative datasets from refs. [2,9,21] (around 2600 BCE), these differences fall within the uncertainty of the analysis (Supplementary Fig. 6), so may be considered suggestive of an earlier arrival of steppe-related ancestry, but not conclusive. We also caution that differences are likely affected by uneven sampling through time in the three different datasets, and so expect the precision of this analysis to improve with denser temporal sampling in the future.

**Timing and duration of the genetic turnover**. We used the software *DATES* (ref. [22], https://github.com/priyamoorjani/DATES) to estimate the admixture time between YAM-related and Late Neolithic ancestries in all Final Neolithic and Early Bronze Age individuals from Switzerland with substantial admixture proportion. Our estimates range between 3 and 60 generations ago, with substantial uncertainty. If the mixture occurred as one single event in the history of all individuals, we would expect more recent individuals to have a higher admixture time estimate (i.e., more generations ago) than individuals of older dating. However, we observe only a slight trend towards more generations in individuals with younger C[14] dates (Fig. 3a), which suggests that the process of admixture with steppe-related ancestry occurred over several hundred years rather than as a single pulse.

For comparison, we performed a similar analysis for published ancient Final Neolithic and Bell Beaker populations from the MES region in Germany, Great Britain, and Iberia using a mixture between the YAM-related steppe component and the corresponding Middle Neolithic population of the region (Fig. 3c). For the MES region from Germany, Late Neolithic and Early

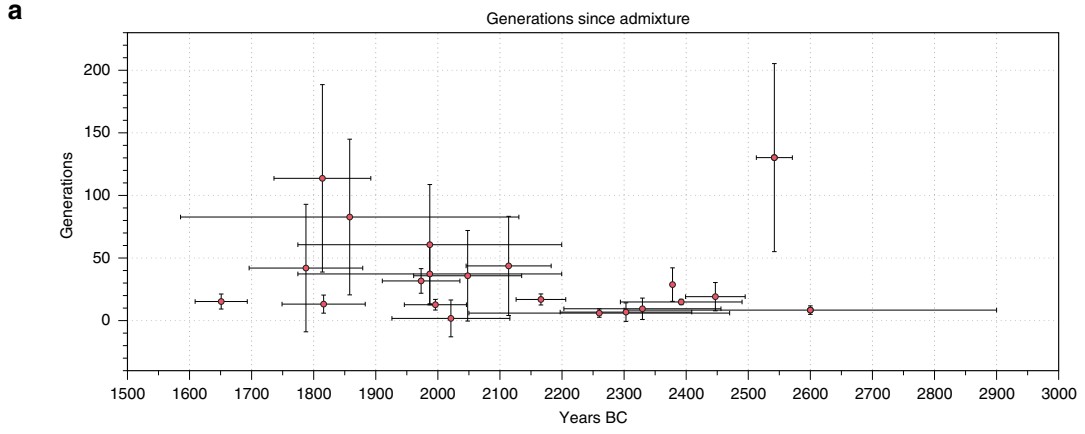

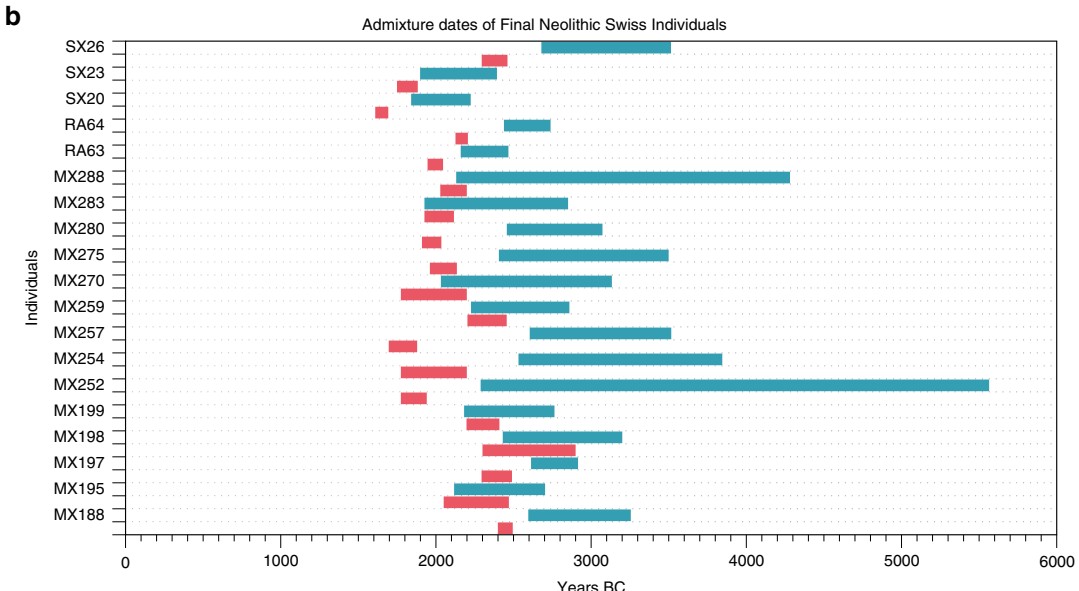

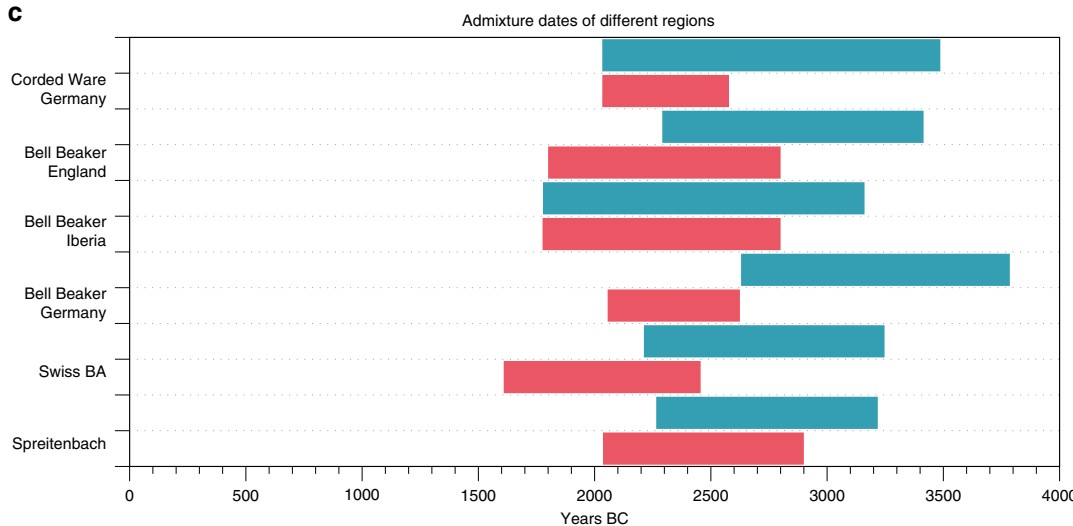

**Fig. 3 Estimated admixture times between the Yamnaya-like steppe component and the Middle Neolithic population using DATES software.**
**a** Admixture dates of single individuals are plotted against their calC14 dates (horizontal error bars indicate uncertainty in C14 dating and vertical error bars show 95% confidence interval of generation times) and **b** displayed as time range (C14 dates in red and estimated admixture dates in turquoise). **c** Admixture dates of grouped individuals according to their regions of origin were calculated (colors as above).

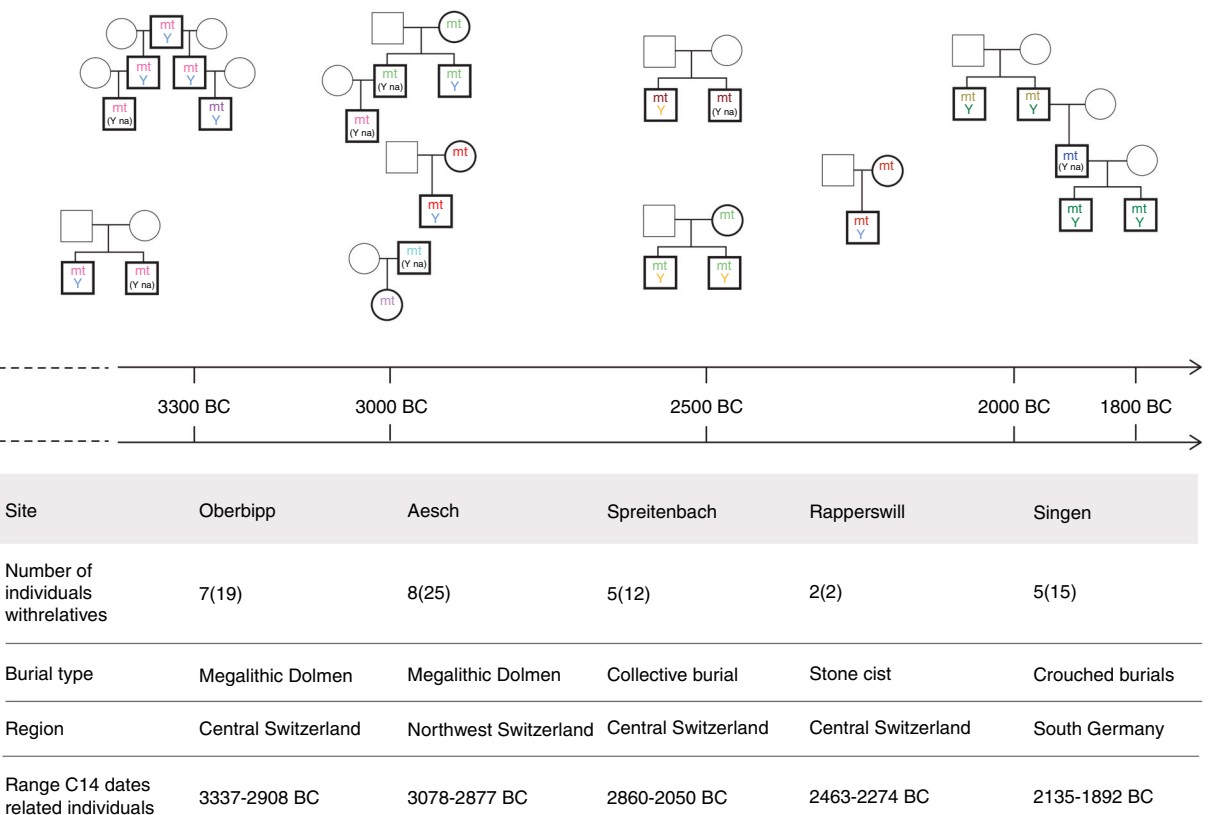

| Site | Oberbipp | Aesch | Spreitenbach | Rapperswill | Singen |
|---|---|---|---|---|---|
| Number of individuals with relatives | 7(19) | 8(25) | 5(12) | 2(2) | 5(15) |
| Burial type | Megalithic Dolmen | Megalithic Dolmen | Collective burial | Stone cist | Crouched burials |
| Region | Central Switzerland | Northwest Switzerland | Central Switzerland | Central Switzerland | South Germany |
| Range C14 dates related individuals | 3337-2908 BC | 3078-2877 BC | 2860-2050 BC | 2463-2274 BC | 2135-1892 BC |

**Fig. 4 Reconstructed family trees from different sites over time.** All relationships between the single individuals were reconstructed from autosomal variants and confirmed by uniparentally inherited markers such as mtDNA haplotype and Y chromosomal haplogroup. Individuals with black outline were available for analysis and individuals with gray outline were not found within the burials and are missing. Same colors indicate identical mtDNA haplotypes and matching Y chromosomal haplogroups.

Bronze Age individuals were split in Bell Beaker and Corded Ware groups. Similar to this approach, we also split the Neolithic individuals from Switzerland into a group of individuals associated with the Corded Ware Complex from Spreitenbach and a group of younger individuals from the Bronze Age. All four regions (Iberia, Great Britain, MES and Switzerland) show similar ranges of admixture dates between the steppe-related component and the Neolithic component starting between 3000 and 2500 BCE.

The inferred admixture time describes the time when people of steppe-related ancestry encountered people with Late Neolithic ancestry but does not reveal the location. Admixture could have happened in Switzerland or elsewhere, with already admixed individuals moving to Switzerland.

**Neolithic source population for the admixing event**. We tested which Neolithic population likely admixed with the incoming people that carried large amounts of YAM-related ancestry, by adding Late Neolithic individuals from Switzerland, Globular Amphora, Iberia Middle Neolithic, and France Neolithic groups separately as additional right populations to the three-way model (WHG, Western Anatolia Neolithic and Yamnaya Samara) used in the above qpAdm analyses (ref. [19], Supplementary Data 3 and 4). The model remains fitting for Iberia Middle Neolithic and France Neolithic populations ($p \geq 0.01$ and $p \geq 0.3$, respectively) but it fails when we add Late Neolithic individuals from Switzerland or Globular Amphora as additional right population ($p \leq 2e{-}7$ and $p \leq 2e{-}6$, respectively). This suggests that both the local Swiss Late Neolithic population as well as people associated with the Late Neolithic Globular Amphora culture, located further east, are better proxies for the genetic sources for Final

Neolithic and Bronze Age populations from Switzerland than Western ANF and steppe pastoralists.

**Kinship before and after the genetic transition**. In five burial sites, we identified first-degree relatives using the software lcmlkin[23] and READ[24], and by calculating pairwise mismatch rates across all analyzed genomic sites between individuals (see "Methods"). Four of these sites contained more than two closely related individuals, which allowed us to reconstruct family trees spanning three generations for Oberbipp, Aesch and Singen (Fig. 4). In these multiple burials, only a few female individuals (four individuals) were buried together with one of their parents or their sons, compared with a higher number (21 individuals) of males buried with their father, brothers or sons, indicating that males likely tended to stay where they were born, while females were likely mobile. This pattern is observed both before and after the arrival of the YAM-related ancestry and is indicative of patrilocal societies during Late Neolithic times in the studied region, consistent with previous results from Neolithic times throughout Northern and Western Europe[25,26].

**Population movement in Switzerland after the Bronze Age**. We compared present-day Swiss people to regional Final Neolithic populations (Spreitenbach, Bad Zurzach, Wartau) to test whether there are additional ancestry components in present-day Swiss people. For that analysis, we made use of information available in the present-day dataset (POPRES) about the self-reported language group and split the individuals into three linguistic regions, as they were shown to be distinguishable genetically in previous studies[27]: German-speaking, French-speaking, and

Italian-speaking Switzerland. To test for continuity between the ancient and present-day population we used the method *qpWave* from the ADMIXTOOLS package[19] and found that a simple continuity can be rejected ($p = 0.0003$) for all three linguistic regions separately and the entire present-day Swiss population combined, consistent also with the PCA (Fig. 1c).

To assess whether ancient Swiss individuals from the Final Neolithic are symmetrically related to different linguistic present-day groups, or share an excess of alleles with any of them, we calculated D-statistics of the form D(Mbuti, Test, Swiss–French, Swiss–Italian), D(Mbuti, Test, Swiss–German, Swiss–Italian), and D(Mbuti, Test, Swiss–French, Swiss–German) where "Test" are the different Neolithic groups. The first two D-statistics were, with few exceptions, all negative with a $|Z| \geq 1.099$ (maximum values to be found in D(Mbuti, Test, Swiss–French, Swiss–Italian) for Singen with $-3.431$ and Bad Zurzach $-3.068$) indicating the least genetic affinity of the Final Neolithic and Early Bronze Age individuals of this study to the Italian-speaking group in our present-day Swiss dataset. For D(Mbuti, Test, Swiss–French, Swiss–German) some variation between the sites can be found (Fig. 5) with the older sites sharing more alleles with the French-speaking group, and the younger sites being more similar with the German-speaking group.

**Analysis of functional SNPs.** We analyzed the frequencies of several phenotypic SNPs (Table 1, "Methods"). Derived alleles for SLC24A5 associated with light skin pigmentation in Europeans were found in all individuals with this position covered. The frequency of SLC45A2 also causing lighter skin pigmentation tends to increase and the frequency of HERC2 associated with light eye-color tends to decrease towards the Final Neolithic. A mutation associated with lactose tolerance in adulthood (LCT; rs4988235), which is of high frequency in Europe today, is absent in Late and Middle Neolithic samples. The only exception is one Final Neolithic individual from Spreitenbach dating to 2105–2036 calBCE, which is one of the earliest European individuals with this mutation found so far. The near absence of lactose tolerance in these ancient groups is in concordance with previous studies hypothesizing that this mutation arose in the Final Neolithic period and started to increase in frequency after the beginning of the Bronze Age[2].

## Discussion

Our study is the first to report a substantial number of ancient genomes from Switzerland, following a trend of population-scale archaeogenetic sequencing studies in Europe[9,21,28], made possible by capture technology. In accordance with previous studies[1,6,7], the Middle and Late Neolithic Swiss individuals are descendants of late European hunter-gatherers and early farmers, whilst the individuals after 2700 BCE also carry steppe-related ancestry[6,7,9]. Genetic similarities between Corded Ware associated individuals from the MES region in Germany and individuals from Spreitenbach, also associated with the Corded Ware Complex, suggest that this complex was associated with a relatively homogenous genetic population throughout/across large parts of Central Europe.

The social and family structures, as reconstructed by biological kinship networks, remain the same before and after the arrival of steppe-related ancestry in the region. The predominant social structure in populations buried at the sites investigated in this study must have been a patrilocal society where males stayed where they were born, and females came from more distant living families, a societal dynamic which has been confirmed by stable isotopes[29] and that has been previously documented for the Middle Neolithic[25]. Also, higher female mobility has been shown

during the Early Bronze Age[26,30]. Our study also presents one of the earliest evidence for adult lactose tolerance in Europe, dating to 2105–2036 calBCE.

Unsurprisingly, comparing our ancient individuals from Switzerland with the data of individuals from present-day Switzerland reveals additional changes in the region since the Bronze Age. In the periods following the studied time span, different factors could have influenced the population. In particular, in the so-called migration period from 375 to 538 AD, following the Roman Empire, in which there was widespread migration of peoples within or into Europe[31].

Remarkably, we identified several female individuals without any detectable steppe-related ancestry up to 1000 years after this ancestry arrives in the region, with the most recent woman without such ancestry dating to 2213–2031 calBCE. This suggests a high level of genetic structure in this region at the beginning of the Bronze Age with potential parallel societies living in close proximity to each other. Published stable isotope results for one of these females (MX193 or Individual 3 in the original publication) indicate that she was not of local origin[32]. It can, therefore, be speculated whether admixture between the newly established local population with steppe-related ancestry and mobile females with less or none of it, caused the decline in the relative amount of this ancestry component in the centuries after its arrival in present-day Switzerland. As the parents of those mobile females also could not have carried steppe-related ancestry, it remains to be shown where in Central Europe such populations without this component were present. One possibility could be Alpine valleys, which until today are inhabited by linguistic isolates that exhibit strong genetic differentiation as initial studies on uniparentally inherited markers have suggested[33,34]. But considering the results of Mittnik et al.[35] with similar patterns in the Lech valley, the origin of this steppe-related ancestry component lacking population does not necessarily lie that far south.

Stable isotope analyses[29,32] do not give clear indications if all four females originated and spend their entire life within the region of modern-day Switzerland. Therefore, it cannot be excluded, that these females also originate from regions further south since also some regions e.g., Italy are not genetically described so far for this particular time span. However, individuals without any steppe-related ancestry can be found up until 2479–1945 BCE for example in Iberia or until 2900–1700 BCE in the Minoan population of Crete[21,36] and even later on Sardinia where steppe-related ancestry arrives around 300 CE[37] and where studies of present-day Sardinians found indications of continuity in mountainous regions since Neolithic times[38].

Also noteworthy is the remarkably early arrival of the steppe-related ancestry component in Switzerland, at least as early as or even earlier than in regions of Germany and Great Britain. However, further investigations are needed, especially since datasets from the regions in Great Britain and the MES region in Germany show gaps in the sampling between the Late/Final Neolithic and the Early Bronze Age potentially biasing the results, to draw any conclusions about the exact route in which the steppe-related ancestry spread through Central Europe.

## Methods

**DNA extraction and library preparation**. A total of 263 samples were screened for DNA preservation for this study. A detailed description of the archaeological context of the samples and radiocarbon dates for all individuals included in genome-wide analysis can be found in Supplementary Notes 1 and 2.

All pre-PCR steps were performed in the cleanroom facilities at the Institute for Archaeological Sciences in Tübingen. For the reduction of surface contamination, the samples were treated with UV light for 30 min each side. Between 30 and 50 mg of powder from coronal dentin for teeth and between 50 and 100 mg of bone powder for the petrous bones were used for extraction. After the dissolving of the

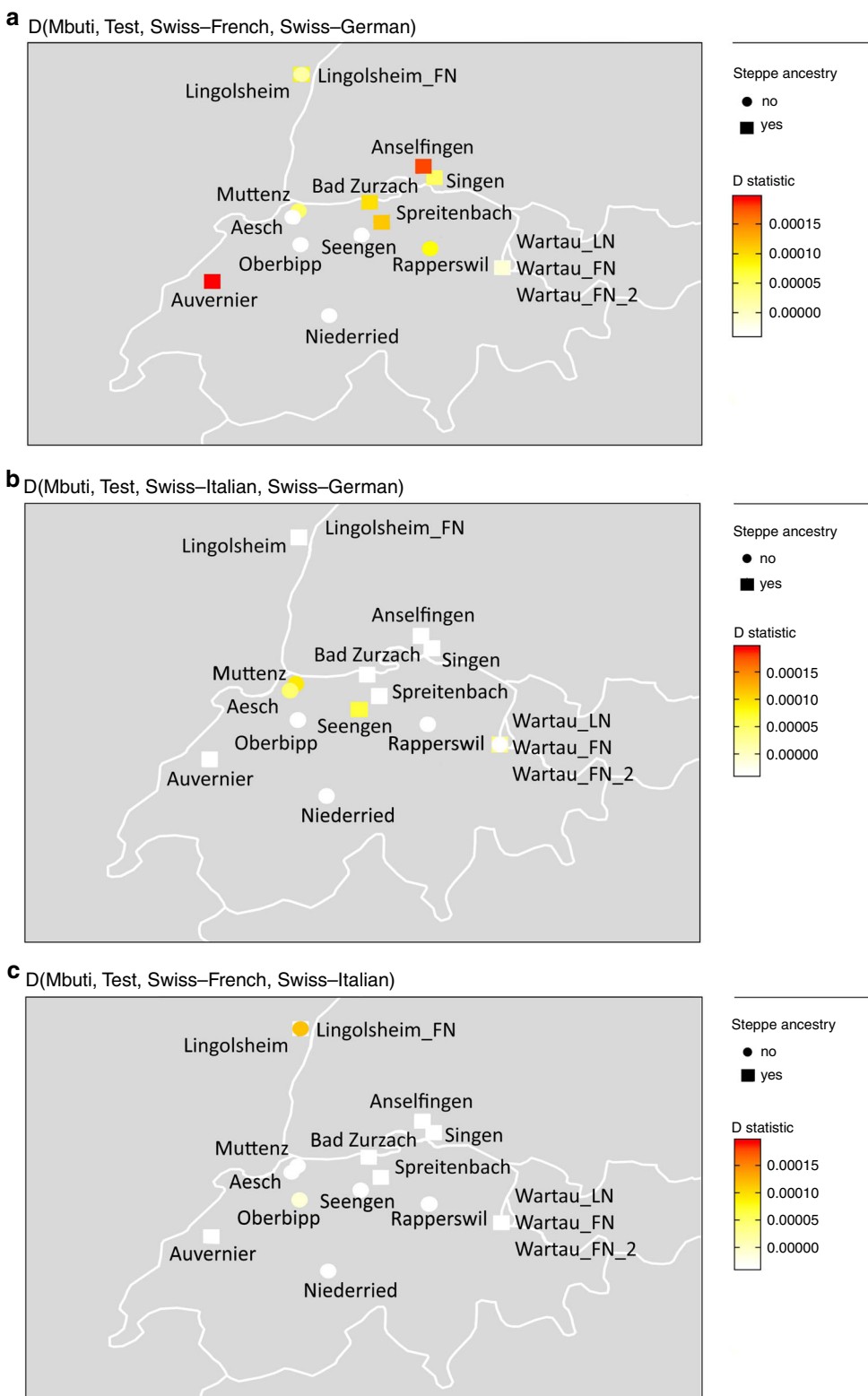

**Fig. 5 Genetic affinity between the ancient Individuals and the French-, German-, and Italian-speaking regions of Switzerland.** Differences in genetic affinity of the ancient individuals from Switzerland to **a** Swiss–French and Swiss–German, **b** Swiss–Italian and Swiss–German, and **c** Swiss–French and Swiss–Italian. Maps generated with R version 3.4.3 (R Core Team 2017) using the CIA World Data Bank II are currently (mid 2003) available from http://www.evl.uic.edu/pape/data/WDB/.

**Table 1 Frequency of the derived allele of four phenotypic SNPs.**

| | Frequency of derived allele SLC45A2 (rs16891982) | Number of individuals position covered SLC45A2 (rs16891982) | Frequency of derived allele SLC24A5 (rs1426654) | Number of individuals position covered SLC24A5 (rs1426654) | Frequency of derived allele HERC2 (rs12913832) | Number of individuals position covered HERC2 (rs12913832) | Frequency of derived allele LCT (rs4988235) | Number of individuals position covered LCT (rs4988235) |
|---|---|---|---|---|---|---|---|---|
| **Late Neolithic** | | | | | | | | |
| Niederried | 0% | 0/2 | 100% | 2/2 | 50% | 2/2 | 0% | 2/2 |
| Oberbipp | 31% | 13/19 | 100% | 7/19 | 50% | 10/19 | 0% | 11/19 |
| Aesch | 57% | 23/25 | 100% | 8/25 | 50% | 18/25 | 0% | 23/25 |
| Muttenz | 0% | 5/5 | 100% | 1/5 | 75% | 4/5 | 0% | 5/5 |
| Seengen | 0% | 0/2 | 0% | 0/2 | 0% | 0/2 | 0% | 0/2 |
| **Final Neolithic** | | | | | | | | |
| Spreitenbach | 55% | 11/12 | 100% | 5/12 | 36% | 10/12 | 8% | 12/12 |
| Bad Zurzach | 0% | 2/2 | 100% | 1/2 | 100% | 2/2 | 0% | 2/2 |
| Singen | 77% | 13/15 | 100% | 4/15 | 13% | 8/15 | 0% | 13/15 |
| Anselfingen | 0% | 1/1 | 0% | 0/1 | 100% | 1/1 | 0% | 1/1 |
| Rapperswil | 100% | 2/2 | 0% | 0/2 | 100% | 1/2 | 0% | 2/2 |
| Auvernier | 0% | 0/1 | 0% | 0/1 | 0% | 0/1 | 0% | 0/1 |
| **Mixed** | | | | | | | | |
| Lingolsheim | 33% | 3/4 | 100% | 1/4 | 0% | 2/4 | 0% | 4/4 |
| Wartau | 83% | 6/8 | 0% | 0/8 | 66% | 4/8 | 16%[a] | 6/8 |

SLC45A2 and SLC24A5 contribute to lighter skin pigmentation, HERC2 is associated with blue eyes and LCT with lactose tolerance in adults.
[a]One individual dating to 789calBCE-2AD.

powder in 1 ml extraction buffer (0.45 M EDTA, 0.25 mg/ml proteinase K) at 37 °C overnight, than the supernatant was transferred into 10 ml binding buffer (5 M GuHCl, 40 % Isopropanol, 115 mM NaAc) and the DNA was bound to a silica membrane in the MinElute Columns from Qiagen. Then the membrane was washed two times with 720 µl of the commercial PE buffer form Qiagen, and then the DNA was eluted in 100 µl TET buffer (1 mM EDTA, 10 mM Tris-HCl, 0.05 % Tween-20, ref. [38]). Sequencing libraries were prepared after[39]. For the blunt ending 20 µl extract were combined with 30 µl reaction mix (1× NEB buffer 2, 100 µM dNTP mix, 0.8 mg/ml BSA, 1 mM ATP, 0.4 U/µl T4 Polynucleotide Kinase (BioLabs, Frankfurt), 0.024 U/µl T4 Polymerase also from BioLabs), and then incubated in a thermo cycler for 15 min at 15 °C and then 15 min at 25 °C. The resulting 18 µl was than used for the ligation of the adapters. With a final volume of 40 µl the reaction contained: 1× Quick Ligase buffer, 250 nM Solexa Adapter Mix and 0.125 U/l Quick Ligase (BioLabs, Frankfurt). The incubation was at room temperature for 20 min. This step was followed by a MinElute purification with an elution volume of 20 µl. The 20 µl from the step before were combined with 20 µl of the reaction mix (1× Isothermal buffer, 125 nM dNTP, 0.4 U/µl Bst polymerase 2.0 from BioLabs), and then incubated for 20 min at 37 °C and then 20 min at 80 °C.

To enable multiplexed sequencing, sample-specific barcodes were added to each library by amplification with tailed primers[40]. The PCR reactions had a final volume of 100 µl and the following concentrations: 1× buffer, 0.25 mM dNTP mix, 0.3 mg/ml BSA, 0.2 mM P7, 0.2 mM P5 and 0.025 U/µl Pfu Turbo Polymerase (Agilent Technologies, Santa Clara, USA). The thermal profile started with 2 min at 95 °C This was then followed by ten cycles with 30 s at 95 °C, 30 s at 58 °C, and 1 min at 72 °C followed by 10 min at 72 °C. After MinElute purification the DNA was eluted in 50 µl TET.

To achieve a high copy number of each library, an additional amplification was performed with Herculase II in a final volume of 100 µl consisting of 1× Herculase II reaction buffer, 0.25 mMdNTPs, 0.4 µM IS 5, 0.4 µM IS 6, 0.01 % Herculase II Fusion DNA Polymerase. The thermal profile started with 2 min at 95 °C followed by a sample-specific number of cycles of 10 s at 95 °C, 30 s at 65 °C, and 30 s at 72 °C. This was then followed by 4 min at 72 °C. All samples were pooled equimolar at 10 nM for shotgun sequencing.

**Mitochondrial and nuclear capture.** In addition to shotgun sequencing, all libraries were enriched for mitochondrial DNA using baits generated from modern DNA via long-range PCR[41]. For extracts with sufficient endogenous DNA after screening two more double-indexed UDG-half treated libraries were prepared similar to the approach described above with the differences that the first master mix consisted of 60 µl with 1× Buffer Tango (Thermo Fisher Scientific), 100 µM dNTPs, 1 mM ATP, and 0.06 U/µl USER enzyme (NEB). After a 30 min incubation at 37 °C, 0.12 U/µl UGI was added and the reaction was incubated for another 30 min at 37 °C. This was followed by the procedure described above[42]. The non-UDG treated library and the two UDG-half libraries were enriched for 1.2 Mio nuclear SNPs using an in-solution hybridization protocol[2,43].

**Bioinformatic processing.** After sequencing all data were processed using the software package EAGER[44]. Adapters were removed and paired-end data were merged using Clip&Merge[44]. Mapping was performed with BWA with the mismatch parameter set to 0.01 and a seed length of 1000 against the human genome reference GRCh37/hg19. If necessary, PCR duplicates were removed using DeDup[44]. BAM files from different libraries of the same extract were merged after

mapping and quality control. Reads mapping to the mitochondrial genome were extracted from the BAM files and the mitochondrial genome was reconstructed and the amount of mitochondrial contamination was estimated using the software *schmutzi*[45]. The web-based tool HaploGrep[46] was used to determine mitochondrial haplogroups. Genetic sex was determined by comparing X chromosomal reads to Y chromosomal reads[47].

For male samples, nuclear contamination was estimated using ANGSD[48]. Samples with mtDNA or nucDNA contamination >5% as well as samples with <10,000 SNPs of the HO dataset were excluded from further analysis.

**Population genetic analysis.** Pseudo-haploid genotypes for the 1.2 M SNPs were retrieved using *pileupcaller* (https://github.com/stschiff/sequenceTools). A reference dataset of 399 published ancient genomes, the HO dataset or the POPRES dataset were compiled. The overlap between the 1.2 M targeted SNPs and the HO data are 593,054 SNPs, between the POPRES and the target SNPs 133,682.

For the principal component analysis on the newly sequenced individuals, published ancient individuals and the European population from the POPRES *smartpca* from the EIGENSOFT package (version: 16000) was used with the parameters *lsqproject: YES* and *shrinkmode: YES*.

Relative proportions of ancestry components in the newly sequenced individuals, published ancient individuals from Germany, Great Britain, and Iberia were estimated using *qpAdm* (version: 632) from ADMIXTOOLS (ref. [19], https://github.com/DReichLab) using a threshold of 100k SNPs for analysis on an individual level (Supplementary Note 3) and modern reference individuals (Mbuti, Papuan, Onge, Han, and Karitiana) from the HO dataset and published ancient individuals (Ust Ishim, Ethiopia 4500BP, Villabruna, MA1).

**Estimation of steppe arrival times.** The arrival time of steppe ancestry was modeled from the newly sequenced individuals and published ancient genomes from Germany, Great Britain, and Iberia assuming that proportions of steppe ancestry increase from zero to $p_e$ at some time $t_e$, and then decrease according to an exponential curve such that they are projected to reach a proportion of $p_0$ at time zero. We used starting values of $p_0 = 0.1$, $p_e = 0.8$, and $t_e$ equal to the mean date of the samples with nonzero steppe ancestry. We then chose optimal parameter values for the model by finding values for $p_0$, $p_e$, and $t_e$ that minimized the residual sum of squares function between fitted and observed steppe ancestry proportions. Analyses were performed using R version 3.4.3 (R Core Team 2017) with the function *optimx*[49]. To avoid downward-biased estimates of $p_e$, we set individuals with steppe ancestry proportion zero, which are observed after $t_e$, to have zero residual value.

To incorporate uncertainty in dating estimates, we did not use the mean date for each individual, and instead randomly sampled a date for each individual from a normal distribution truncated such that the upper and lower bounds of the date estimates form a 95% confidence interval around the sampling mean. We repeated this sampling process 100,000 times and found an optimal value for $t_e$ for each sample, for each dataset. Optimal values of $t_e$ were only retained if they produced stable solutions with a positive-definite Hessian matrix.

For the uncertainty in the proportion of steppe ancestry it is assumed it is identically distributed, with mean zero and unbiased for all points, as it is standard in least-squares regression. We report kernel density estimates weighted by the ratio of the exponential of the sum of the squared residuals, compared with the optimal residual sum of squares observed, which is proportional to the ratio of Gaussian likelihood functions. This approach allows parameter values that

performed best to be given more weight in the report kernel density functions and approximates implementing a Monte Carlo Markov chain with a truncated Gaussian prior distribution for sampling times, a Gaussian distribution for the residuals, and a uniform proposal density function for parameter values.

**Pairwise mismatch rates**. The pairwise mismatch rates were calculated from the genotype file and only pairs with more than 10,000 overlapping SNPs of the 1240k SNP panel were included. Due to the heavy right-skewed distribution of the pairwise mismatch rates, we used a robust linear mixed-effects model to assess the difference between pairwise mismatch rates before and after 2700 BCE. To account for correlations in the data due to the same samples being used to calculate multiple pairwise distances, we included the ID of the samples as random effects. We found that both the time period (before and after 2700 BCE) and the population of origin for samples were significant predictors for the pairwise mismatch rate. Analyses were performed using R version 3.4.3 (R Core Team 2017) via the *robustlmm* package[50].

**Admixture date estimates**. Admixture dates were estimated using DATES, which was extensively tested in ref. [22], based on the 1240k SNP panel for single individuals and for groups of individuals (ref. [22], https://github.com/priyamoorjani/DATES).

**Kinship analysis**. Kinship between individuals was assessed using the software READ[23], lcmlkin[24], and calculating pairwise mismatch rate on autosomal markers and confirmed by mtDNA haplotype and Y chromosomal haplogroups based on the 1240k SNP panel (Table 1).

**qpWave analysis**. Population continuity since the Bronze Age was tested with *qpWave* (version: 410) from ADMIXTOOLS (https://github.com/DReichLab) between newly sequenced Swiss Bronze Age individuals and modern Swiss individuals from POPRES using Mbuti, Karitiana, Hakka Taiwan, Papuan, Onge, Han, Hungarian, MA1, EHG, and WHG as outgroups and differences in the genetic affinity between the Bronze Age individuals and the four linguistic regions of Switzerland were assed with D-statistics of the form $f4(X,Y; Test, Outgroup)$.

**Functional SNP analysis**. Phenotypic SNPs were genotyped using GATK version 3.8[51]. Low coverage positions (e.g., 1×) with reference or alternative alleles carrying A or T were inspected manually to exclude the influence of ancient DNA damage.

## Data availability

Genome-wide data are accessible in the Sequence Read Archive under the accession numbers SAMN14206592–SAMN14206687 (https://www.ncbi.nlm.nih.gov/sra/PRJNA608699) and mitochondrial consensus sequences on GenBank under the accession numbers MT079018–MT079111. The POPRES dataset contains sensitive patient information and is therefore not publicly available but access can be requested via https://www.ncbi.nlm.nih.gov/projects/gap/cgi-bin/study.cgi?study_id=phs000145.v4.p2.

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

# ARTICLE

40. Meyer, M. & Kircher, M. Illumina sequencing library preparation for highly multiplexed target capture and sequencing. *Cold Spring Harb. Protoc.* 5448 (2010).

41. Kircher, M., Sawyer, S. & Meyer, M. Double indexing overcomes inaccuracies in multiplex sequencing on the Illumina platform. *Nucleic Acids Res.* **1**, e3 (2011).

42. Furtwängler, A. et al. Ratio of mitochondrial to nuclear DNA affects contamination estimates in ancient DNA analysis. *Sci Rep.* **8**, 1–8. https://doi.org/10.1038/s41598-018-32083-0.

43. Rohland, N., Harney, E., Mallick, S., Nordenfelt, S. & Reich, D. Partial uracil-DNA-glycosylase treatment for screening of ancient DNA. *Philos. Trans. R. Soc. B* **370**, 20130624 (2014).

44. Fu, Q. et al. DNA analysis of an early modern human from Tianyuan Cave, China. *Proc. Natl Acad. Sci.* **110**, 2223–2227 (2013).

45. Peltzer, A. et al. EAGER: efficient ancient genome reconstruction. *Genome Biol.* **17.1**, 1 (2016).

46. Renaud, G., Slon, V., Duggan, A. T. & Kelso, J. Schmutzi: estimation of contamination and endogenous mitochondrial consensus calling for ancient DNA. *Genome Biol.* **16.1**, 1 (2015).

47. Mittnik, A., Wang, C.-C., Svoboda, J. & Krause, J. A molecular approach to the sexing of the triple burial at the upper paleolithic site of Dolní Věstonice. *PLoS ONE* **11**, e0163019 (2016).

48. Rasmussen, M. et al. An Aboriginal Australian genome reveals separate human dispersals into Asia. *Science* **334**, 94–8 (2011).

49. Nash, J. C. & Varadhan, R. Unifying optimization algorithms to aid software system users: optimx for R. *J. Stat. Softw.* **43**, 1–14, http://www.jstatsoft.org/v43/i09/ (2011).

50. Koller, M. robustlmm: an R package for robust estimation of linear mixed-effects models. *J. Stat. Softw.* **75.6**, 1–24. (2016).

51. McKenna, A. et al. The Genome Analysis Toolkit: a MapReduce framework for analyzing next-generation DNA sequencing data. *Genome Res.* **20**, 1297–1303 (2010).

## Acknowledgements

This study was funded by the German Research Foundation KR 4015/4-1 and the Swiss National Foundation CR31I3L_157024. This project has also received funding from the European Research Council (ERC) under the European Union's Horizon 2020 research and innovation program under grant agreement no. 771234—PALEoRIDER (W.H., A.B.R., and L.P.) and the Heidelberg Academy of Sciences (WIN project "Times of Upheaval: Changes of Society and Landscape at the Beginning of the Bronze Age") (P.W.S. and J.K.). We thank Anna Sapfo Malaspinas, Ivan P. Levkivskyi, Sven Bergman, Michal Feldman, and Choongwon Jeong for helpful discussions and comments. Special thank goes to the Archaeological Service of the Canton of Bern (Switzerland). The authors also would like to thank the Archaeological Services of the Cantons of Aargau (Switzerland), Basel-Landschaft (Switzerland), Solothurn (Switzerland), St. Gallen (Switzerland), and the Archaeological Services of Baden-Württemberg (Germany) and Alsace (France). Furthermore, we thank the Swiss National Museum of Zurich and Sönke Szidat and his team from the LARA laboratory at Bern University for radiocarbon dating and fruitful discussions.

## Author Contributions

A.H., S.L., and J.K. conceived of the study. A.F., E.R., and G.U.N. performed laboratory work. A.F., A.B.R., T.C.L., L.P., W.H., and S.S. analyzed data. I.S. and S.L. performed anthropological assessments. I.S., N.S., J.H., A.D., B.S., J.W., P.W.S., S.L., and A.H. assembled and interpreted archaeological material. A.F., A.B.R., and S.S. designed figures. V.J.S. and J.K. supervised laboratory work. A.F., S.S., and J.K. wrote the paper with input from all co-authors.

## Competing interests

The authors declare no competing interests.
