## [Peer Review File · Nature Communications]

Reviewers' Comments:

Reviewer #1:

Remarks to the Author:

Furtwängler et al present the first in-depth archaeogenomic study that focuses on the geographic area of modern-day Switzerland and the processes around the Late Neolithic period. This study follows a series of similar studies focusing on different time periods and regions around Europe. The 97 newly sequenced individuals represent a substantial advancement especially since this particular region has not been strongly represented in the archaeogenomic record. The main results are similar to what is seen in other parts of Europe at this time period (a major population transformation during the Late Neolithic/Bronze Age) which could have been expected. The main surprise is that the transition appears less abrupt than in other parts of Europe (but see my comments below) which may be due to the specifics of this geographic region. I do not see any major issues with this manuscript but it is rather short so I would like to ask the authors to add some additional information for better understanding. I am outlining more specific comments below:

===Results/Discussion===

- The results section does not include any information on uniparental markers or contamination estimates/authentication.
- Dating the arrival of Yamnaya ancestry and the conclusions that the turnover has been complex depends on the reliability of radiocarbon dates for the new individuals. Finding individuals without steppe ancestry 1000 years after the arrival of that ancestry seems surprising, so the radiocarbon dates of those should be really solid. I was not able to find much information about whether all individuals are dated directly or contextually (see below). Furthermore, I am wondering if other effects like calibration and potential reservoir effects could have impacted this. Considering that the introduction mentions the CWC sites are along the big lakes, there is a potential that their diet was based on resources from that lake which may bias radiocarbon dates. If available, the authors should add information on the dietary behavior of these individuals to show that such potential biases did not affect their conclusions.
- Related to the observation that the four individuals with no steppe ancestry are all females. Do the authors see evidence for sex biased admixture processes? Comparing the X chromosome to the autosomes could give some indication. Saag et al (2017) and Goldberg et al (2016) have done similar analyses for other parts of Europe. The sample size in this study should suffice to provide such a perspective. This could also give insights into potential social structure as proposed by the authors.
- The authors cite studies from Iberia (Olalde et al 2018, 2019) but Iberia is not included in the analysis shown in Fig 2. I would think that Iberia has been sampled quite densely and I suspect the arrival of steppe ancestry to the peninsula has been quite complex as well. A comparison between Iberia and Switzerland would be quite interesting.
- The manuscript contains an interesting comparison between the ancient groups and present-day Swiss. The authors could elaborate a bit more on this part and draw conclusions from the results connecting the observations to historical and prehistoric processes.

===Materials and Methods===

Just reading the article, I was not able to follow the entire procedure conducted to achieve these results. Some additional information can be found in figure/table legends (e.g. in the supplementary Excel sheet) while others can only be guessed. Please provide additional information to make this study reproducible.

- Are all new samples in this study radiocarbon dated? If yes, please describe how!
- Please cite publications for all softwares used and give version numbers!
- Please include parameters used for computational analyses.
- I am not sure I fully understand what types of analysis are based on the HO SNP panel, the 1240k capture and the POPRES data.
- Why was no unsupervised clustering approach (e.g. ADMIXTURE, Ohana, ngsadmix) used? The authors move from PCA to supervised analysis which may miss other ancestries (although they seem unlikely in this case).
- The pairwise mismatch rate is a nice way to study genetic diversity. It is fundamentally similar to the conditional nucleotide diversity used by previous studies, so please cite accordingly.

- Please give additional information on the qpadm/qpWave analysis. E.g. what outgroups were used?
- Parts of the conclusions are based on the new method DATES. The authors only cite a github address, since no corresponding publication (neither peer-reviewed or biorxiv) was available at the time of submission which makes it difficult to assess whether such methods are tested and appropriate. A description of that method became available just now the supplement of another article (which is not cited as submitted or in press), but I don't think this is good practice.

===Figures===

Fig 1C: The data points almost seem to be arranged on a grid suggesting that some resolution in the coordinates has been lost. Is there a reason for that?

Fig 1C: would it be possible to use one color or symbol shape per time period?

Fig 2B: The sampling for GB and Germany shows much larger gaps than for Switzerland making this comparison difficult.

Fig 2B: I don't think I understand the density curve (if it is one) in this figure. There is no description in the figure legend.

Figs 2 and 3: Could the same colors and symbol shapes as in Fig 1 be used to make it possible to distinguish sites and context?

Reviewer #2:

Remarks to the Author:

This is an important paper that follows a current trajectory in the ancient DNA field towards understanding more localized population dynamics. For a number of years, we have been drawing big arrows on the map, and now it's time to fill in all the gaps and look at the finer details of these prehistoric migration events. I think this paper does an excellent job and I have very few comments. The methods represent standard pop gen tools implying that this study is not driven by technical novelty but rather by characterizing a previously unexplored part of the past European gene pool. I agree on the interpretations of the data and the conclusions make sense in light of what we now know from previous studies of other regions.

I would have liked to see a bit more discussion on what could be called "basic results", like endogenous DNA content, C-T deamination damage levels, contamination levels, how did the dentine samples perform in comparison with the petrous bones, how well did the mtDNA-capture perform etc. etc.? All this is crucial information for evaluating the basic authenticity of the data and the efficiency of the methods, and this is what differentiates this discipline from "standard pop gen" studies. So, a few more words on this, including references to the relevant supplement tables would be great. Apart from this, I think this is a great study with a high technical standard and with some important results that are suited for publication in Nature Communications.

cheers,

Morten Allentoft

Reviewer #3:

Remarks to the Author:

This is a well-written and interesting paper on the arrival of steppe ancestry in Switzerland. I think it can be published with some revisions. My main concern is on the central interpretation of "parallel societies" that is based on a handful of late female individuals and could have alternative interpretations (see below).

Paper:

- Page numbers would be helpful.

34: as early as: it would be better I think to replace the broad range with the range of the earliest individual with steppe ancestry.

41: Western Anatolia. This is uncertain, as the Western Anatolia-like farmers extended into Central

Anatolia and also Southeastern Europe. I don't think one can claim that the first farmers of Europe came from Western Anatolia specifically (at least not on the basis of the genetic data).

84-102: Cite Haak et al. (2015) and any other later observations for the observation of increase of hunter-gatherer ancestry and the decrease of steppe ancestry, as this is not unique to Switzerland.

124: Maybe add that this does not necessarily mean local persistence of unadmixed steppe-less populations. The four women could be migrants. (For one of them stable isotopes suggest she was not local). The idea of "parallel societies" which is also used in the title of the paper is interesting, but I don't think fully supported. There are, as I can see it, two possibilities:

- Parallel societies persisting for centuries after the arrival of steppe ancestry so that steppe-less women could be sampled long after the arrival of this ancestry
- Generalized mixture (no parallel societies) coupled with migration from elsewhere (e.g., Italy or Southern Europe) where steppe-less individuals may have persisted longer.

You can perhaps cite work on Iberia, Greece where steppe-less individual persist for a very long time after the first arrival of steppe ancestry in Europe. Where are the last currently known steppe-less individuals in different parts of Europe? In general, I think it would be good to list both possibilities: that of local Central European parallel societies, but also that of migration within Europe.

125: This could be an effect of ascertainment of SNPs on the 1.2M array in present-day humans (?) Demography might also affect this quantity. I don't think it can be easily interpreted in terms of heterogeneity of population. Minimally add some caution on interpretation and modern populations in Fig. 2C

Fig.1C: Consider increasing numerical precision to get rid of digital effect

Fig.2: Make panel B a little taller (too many overlapping bars)

Supplement

34: 13 remaining add space

212: all repeated twice

Table 1: add terminal derived mutation, as Haplogroup names evolve and it will be nice for future reader to know what SNP was derived without tracking down the version of the tree you used

221: "if necessary" please clarify

Supplementary Figure 1 is a little busy with too many overlapping bars. You could make it taller to fill a page and be more legible

SI4: Earliest increase in Switzerland. Per supplementary figure 1, the Swiss transect is fuller than the German one in the critical half-millennium 3000-2500BC. So, add caution that the earliest detection in Switzerland could be due to this and not due to an earlier arrival of the ancestry in Switzerland.

Reviewers' comments:

Reviewer #1 (Remarks to the Author):

Furtwängler et al present the first in-depth archaeogenomic study that focuses on the geographic area of modern-day Switzerland and the processes around the Late Neolithic period. This study follows a series of similar studies focusing on different time periods and regions around Europe. The 97 newly sequenced individuals represent a substantial advancement especially since this particular region has not been strongly represented in the archaeogenomic record. The main results are similar to what is seen in other parts of Europe at this time period (a major population transformation during the Late Neolithic/Bronze Age) which could have been expected. The main surprise is that the transition appears less abrupt than in other parts of Europe (but see my comments below) which may be due to the specifics of this geographic region. I do not see any major issues with this manuscript but it is rather short so I would like to ask the authors to add some additional information for better

understanding. I am outlining more specific comments below:

===Results/Discussion===

- The results section does not include any information on uniparental markers or contamination estimates/authentication.

Some general information about the contamination thresholds used are now mentioned at the beginning of the result section as well as some information about the main results of the analysis of mtDNA.

- Dating the arrival of Yamnaya ancestry and the conclusions that the turnover has been complex depends on the reliability of radiocarbon dates for the new individuals. Finding individuals without steppe ancestry 1000 years after the arrival of that ancestry seems surprising, so the radiocarbon dates of those should be really solid. I was not able to find much information about whether all individuals are dated directly or contextually (see below). Furthermore, I am wondering if other effects like calibration and potential reservoir effects could have impacted this. Considering that the introduction mentions the CWC sites are along the big lakes, there is a potential that their diet was based on resources from that lake which may bias radiocarbon dates. If available, the authors should add information on the dietary behavior of these individuals to show that such potential biases did not affect their conclusions.

It is now mentioned in the results, that every sample is individually radiocarbon dated. Comments on the mentioned biases are in an additional SI section (SI 2) about the radiocarbon dating procedure.

- Related to the observation that the four individuals with no steppe ancestry are all females. Do the authors see evidence for sex biased admixture processes? Comparing the X chromosome to the autosomes could give some indication. Saag et al (2017) and Goldberg et al (2016) have done similar analyses for other parts of Europe. The sample size in this study should suffice to provide such a perspective. This could also give insights into potential social structure as proposed by the authors.

Similar to the approach in Saag et al. f3 values for autosomes and X chromosome were compared and showed as expected some asymmetry with stronger effects of 'steppe'-ancestry on the male side. This analysis is now included in the SI section 6 and in SI Figure 5. It is also now more clearly mentioned in the text: line 155-158.

- The authors cite studies from Iberia (Olalde et al 2018, 2019) but Iberia is not included in the analysis shown in Fig 2. I would think that Iberia has been sampled quite densely and I suspect the

arrival of steppe ancestry to the peninsula has been quite complex as well. A comparison between Iberia and Switzerland would be quite interesting.

The individuals from Olalde et al. 2019 are now included, and added to Figure 2, SI Figure 6 and SI Figure 7.

- The manuscript contains an interesting comparison between the ancient groups and present-day Swiss. The authors could elaborate a bit more on this part and draw conclusions from the results connecting the observations to historical and prehistoric processes.

A small paragraph discussing the population continuity is added to the discussion.

===Materials and Methods===

Just reading the article, I was not able to follow the entire procedure conducted to achieve these results. Some additional information can be found in figure/table legends (e.g. in the supplementary Excel sheet) while others can only be guessed. Please provide additional information to make this study reproducible.

- Are all new samples in this study radiocarbon dated? If yes, please describe how!

Yes. This is now detailed in SI section 2

- Please cite publications for all softwares used and give version numbers!
- Please include parameters used for computational analyses.

Citations and parameters are added if possible and necessary

- I am not sure I fully understand what types of analysis are based on the HO SNP panel, the 1240k capture and the POPRES data.

In the method section of each analysis it is now mentioned on which dataset it is performed.

- Why was no unsupervised clustering approach (e.g. ADMIXTURE, Ohana, ngsadmix) used? The authors move from PCA to supervised analysis which may miss other ancestries (although they seem unlikely in this case).

Results of unsupervised ADMIXTURE supporting the results from the PCA are now added to SI section 5. The results are congruent with previous analyses from Europe and confirm our other analyses.

- The pairwise mismatch rate is a nice way to study genetic diversity. It is fundamentally similar to the conditional nucleotide diversity used by previous studies, so please cite accordingly.

Skoglund et al. 2014 is now cited which describes an analysis with conditional nucleotide diversity.

- Please give additional information on the qpadm/qpWave analysis. E.g. what outgroups were used?

The outgroups are now added, and the analysis is detailed in Methods.

- Parts of the conclusions are based on the new method DATES. The authors only cite a github address, since no corresponding publication (neither peer-reviewed or biorxiv) was available at the time of submission which makes it difficult to assess whether such methods are tested and appropriate. A description of that method became available just now the supplement of another article (which is not cited as submitted or in press), but I don't think this is good practice.

Narasimhan et al. 2019 is now published and cited.

===Figures===

Fig 1C: The data points almost seem to be arranged on a grid suggesting that some resolution in the coordinates has been lost. Is there a reason for that?

Smartpca rounds the eigenvectors to the closest 4th decimal. Additionally, smartpca's "shrinkmode" was used an option that scales all eigenvectors in some way, making all values considerably smaller. This makes the rounding become obvious when plotting. We find this acceptable, given that none of our results is affected by this rounding effect.

Fig 1C: would it be possible to use one color or symbol shape per time period?

Yes. One symbol per time period is now used also in Fig 5.

Fig 2B: The sampling for GB and Germany shows much larger gaps than for Switzerland making this comparison difficult.

We agree. We now make this caveat clearer in the interpretation in the main text and the Discussion.

Fig 2B: I don't think I understand the density curve (if it is one) in this figure. There is no description in the figure legend.

A more detailed description of the curve is now added to the Figure legend.

Figs 2 and 3: Could the same colors and symbol shapes as in Fig 1 be used to make it possible to distinguish sites and context?

Yes. While in Figure 2 and 3 time periods and context are not displayed, the symbols in Figure 5 are now adopted to the symbols used in Figure 1 to distinguish time periods.

Reviewer #2 (Remarks to the Author):

This is an important paper that follows a current trajectory in the ancient DNA field towards understanding more localized population dynamics. For a number of years, we have been drawing big arrows on the map, and now it's time to fill in all the gaps and look at the finer details of these prehistoric migration events. I think this paper does an excellent job and I have very few comments. The methods represent standard pop gen tools implying that this study is not driven by technical novelty but rather by characterizing a previously unexplored part of the past European gene pool. I agree on the interpretations of the data and the conclusions make sense in light of what we now know from previous studies of other regions.

I would have liked to see a bit more discussion on what could be called "basic results", like endogenous DNA content, C-T deamination damage levels, contamination levels, how did the dentine samples perform in comparison with the petrous bones, how well did the mtDNA-capture perform etc. etc.? All this is crucial information for evaluating the basic authenticity of the data and the efficiency of the methods, and this is what differentiates this discipline from "standard pop gen" studies. So, a few more words on this, including references to the relevant supplement tables would be great. Apart from this, I think this is a great study with a high technical standard and with some important results that are suited for publication in Nature Communications.

cheers,
Morten Allentoft

Comments on the contamination estimates are now in the results section of the main text and information about deamination levels, endogenous DNA and duplication rate of the libraries are now in SI Table 2. We also now list the source tissue making it possible to compare Femora, petrous bones and Teeth.

Reviewer #3 (Remarks to the Author):

This is a well-written and interesting paper on the arrival of steppe ancestry in Switzerland. I think it can be published with some revisions. My main concern is on the central interpretation of "parallel societies" that is based on a handful of late female individuals and could have alternative interpretations (see below).

Paper:

- Page numbers would be helpful.

We have added page numbers.

34: as early as: it would be better I think to replace the broad range with the range of the earliest individual with steppe ancestry.

The range is now replaced with the radiocarbon dates of the earliest individual with steppe ancestry. Due to the plateau in the calibration curve the range is not much smaller now.

41: Western Anatolia. This is uncertain, as the Western Anatolia-like farmers extended into Central Anatolia and also Southeastern Europe. I don't think one can claim that the first farmers of Europe came from Western Anatolia specifically (at least not on the basis of the genetic data).

It is now changed to "new incoming people with ancestry related to Western Anatolian early farmers."

84-102: Cite Haak et al. (2015) and any other later observations for the observation of increase of hunter-gatherer ancestry and the decrease of steppe ancestry, as this is not unique to Switzerland.

Haak et al. 2015 and Allentoft et al. 2015 are cited

124: Maybe add that this does not necessarily mean local persistence of unadmixed steppe-less populations. The four women could be migrants. (For one of them stable isotopes suggest she was not local). The idea of "parallel societies" which is also used in the title of the paper is interesting, but I don't think fully supported. There are, as I can see it, two possibilities:

- Parallel societies persisting for centuries after the arrival of steppe ancestry so that steppe-less women could be sampled long after the arrival of this ancestry

- Generalized mixture (no parallel societies) coupled with migration from elsewhere (e.g., Italy or Southern Europe) where steppe-less individuals may have persisted longer.

You can perhaps cite work on Iberia, Greece where steppe-less individuals persist for a very long time after the first arrival of steppe ancestry in Europe. Where are the last currently known steppe-less individuals in different parts of Europe? In general, I think it would be good to list both possibilities: that of local Central European parallel societies, but also that of migration within Europe.

The discussion paragraph is about the four females is extended and the possibility of migration is now mentioned. We also changed the title to: Ancient genomes reveal social and genetic structure of Late Neolithic Switzerland

125: This could be an effect of ascertainment of SNPs on the 1.2M array in present-day humans (?) Demography might also affect this quantity. I don't think it can be easily interpreted in terms of heterogeneity of population. Minimally add some caution on interpretation and modern populations in Fig. 2C

A bias due to demography cannot be excluded completely. However, close relatives are excluded from the analysis and in both groups the majority of individuals originate from multiple burials where distantly related individuals could be expected. Since modern individuals sampled for example the HO dataset usually do not have such similar context comparison between modern populations to our two ancient Swiss groups is problematic. In addition, modern populations were also separated by the ancient groups by several thousands of years of population history and furthermore globalisation heavily increasing diversity in modern groups. However, adding some modern individuals from France and Germany in the statistics (even without correcting for the diploid nature of the data and therefore with extreme bias towards high values) the comparison between the two ancient groups is still significant.

Fig.1C: Consider increasing numerical precision to get rid of digital effect

The PCA plot does not have more resolution due to the low number of SNPs shared between the datasets used for calculation. See also our response to reviewer 1 about this.

Fig.2: Make panel B a little taller (too many overlapping bars)

The size is now increased but many of the individuals just plot very close to each other.

Supplement

34: 13 remaining add space is added

212: all repeated twice Did not find something which is repeated

Table 1: add terminal derived mutation, as Haplogroup names evolve and it will be nice for future reader to know what SNP was derived without tracking down the version of the tree you used

The terminal derived mutations are added to the table

221: "if necessary" please clarify

Is now clarified

Supplementary Figure 1 is a little busy with too many overlapping bars. You could make it taller to fill a page and be more legible

The Figure is now stretched on one entire page. But due to similar dates and proportions many individuals plot close to each other.

SI4: Earliest increase in Switzerland. Per supplementary figure 1, the Swiss transect is fuller than the German one in the critical half-millennium 3000-2500BC. So, add caution that the earliest detection in Switzerland could be due to this and not due to an earlier arrival of the ancestry in Switzerland.

In the interpretation in the main text it is mentioned that these gaps could bias the results.

Reviewers' Comments:

Reviewer #1:

Remarks to the Author:

I thank the authors for their careful revision of the manuscript. This will become a valuable addition to the literature filling some important geographic and temporal gaps. I still have some minor comments but it should be straightforward to address them without an additional round of review.

The text is highlighting some female individuals with zero percent steppe ancestry even 1000 years after the arrival of that component. Figure 2B shows that something similar might have been the case in Iberia where several individuals also show 0% steppe ancestry about 1000 years after the estimated arrival. Figure S6 reveals that those are males. I am actually wondering if it would be a good idea to include sex in 2B as in S6. One should also be more explicit that 2B does not show uncertainties in the steppe ancestry estimates and that those uncertainties were also not used for the curve fitting (as opposed to the uncertainties in radiocarbon dating).

To return to the resolution of figure 1C (as also highlighted by reviewer 3). I agree that none of the conclusions are affected by this merely cosmetic effect but something still seems to be going on. I have seen several smartpca plots in the literature (also using shrinkmode) and I cannot remember such an effect. Are the authors using some additional scaling or non-default parameters for smartpca?

Reviewer #2:

Remarks to the Author:

I think the authors have responded well to the points raised by the three reviewers. In my view the study can now be published.

Reviewer #3:

Remarks to the Author:

The authors have addressed most of my review comments.

The abstract should be edited to supplement the idea of "parallel societies" with the idea that the genetically differentiated individuals could be migrants from southern Europe. The data does reveal the presence of genetically differentiated (with respect to steppe ancestry) individuals in Switzerland but does NOT show that this is due to parallel societies any more than that they were migrants. Both are possible and this should be clearly reflected, not only in the amended title and discussion, but also in the abstract.

In the same vein, one could also add a reference to ancient DNA from Minoan Crete (Lazaridis et al. Nature 2017) and ancient Sardinia (Fernandes et al. bioRxiv 2019) in paragraph 308-314 to supplement the observation on Iberia (Olalde et al. Science 2019). Based on these three studies, steppe-less individuals existed in several geographically distant sampled southern European locations, making the presence of such in Switzerland as migrants from southern Europe a strong possibility, no less stronger than the idea of "parallel societies".

Other than the above, the paper can be published as is.

Reviewer #1 (Remarks to the Author):

I thank the authors for their careful revision of the manuscript. This will become a valuable addition to the literature filling some important geographic and temporal gaps. I still have some minor comments but it should be straightforward to address them without an additional round of review.

The text is highlighting some female individuals with zero percent steppe ancestry even 1000 years after the arrival of that component. Figure 2B shows that something similar might have been the case in Iberia where several individuals also show 0% steppe ancestry about 1000 years after the estimated arrival. Figure S6 reveals that those are males. I am actually wondering if it would be a good idea to include sex in 2B as in S6. One should also be more explicit that 2B does not show uncertainties in the steppe ancestry estimates and that those uncertainties were also not used for the curve fitting (as opposed to the uncertainties in radiocarbon dating).

Sex is now included in figure 2B. Uncertainty in the proportion of Steppe ancestry is not assumed to be absent, but rather is assumed to be identically distributed, with mean zero, and unbiased for all points, as is standard in least-squares regression. We believe these assumptions are reasonable in this context from the visual inspection of a random selection of residual plots, and given that the dominant uncertainty in the final estimates is likely to come from uncertainty in the C14 ages. We have clarified this in the Methods.

To return to the resolution of figure 1C (as also highlighted by reviewer 3). I agree that none of the conclusions are affected by this merely cosmetic effect but something still seems to be going on. I have seen several smartpca plots in the literature (also using shrinkmode) and I cannot remember such an effect. Are the authors using some additional scaling or non-default parameters for smartpca?

No additional scaling or parameters were used. In order to correctly project individuals on a PCA, shrinkage correction is required, which will ensure that the projected individuals are not projected closer to the origin of the PCA plot than they should be. Smartpca provides two ways to correct for shrinkage, with 'shrinkmode: YES' being suggested as a better correction, albeit more computationally intensive.

[See <https://reich.hms.harvard.edu/sites/reich.hms.harvard.edu/files/inline-files/shrinkdemo.tar.gz> for details]

However, when shrinkmode is used, the resulting eigenvectors become smaller while retaining their relative distances. The grid-like pattern of the plot is, therefore, not the result of lower resolution, but an artefact of the precision with which smartpca reports the resulting eigenvectors.

Reviewer #2 (Remarks to the Author):

I think the authors have responded well to the points raised by the three reviewers. In my view the study can now be published.

Reviewer #3 (Remarks to the Author):

The authors have addressed most of my review comments.

The abstract should be edited to supplement the idea of "parallel societies" with the idea that the genetically differentiated individuals could be migrants from southern Europe. The data does reveal the presence of genetically differentiated (with respect to steppe ancestry) individuals in Switzerland but does NOT show that this is due to parallel societies any more than that they were migrants. Both are possible and this should be clearly reflected, not only in the amended title and discussion, but also in the abstract.

Since the word count of the abstract is limited there is no space for both possibilities therefore neither of them is included in the abstract now.

In the same vein, one could also add a reference to ancient DNA from Minoan Crete (Lazaridis et al. Nature 2017) and ancient Sardinia (Fernandes et al. bioRxiv 2019) in paragraph 308-314 to supplement the observation on Iberia (Olalde et al. Science 2019). Based on these three studies, steppe-less individuals existed in several geographically distant sampled southern European locations, making the presence of such in Switzerland as migrants from southern Europe a strong possibility, no less stronger than the idea of "parallel societies".

The mentioned references are now cited in the paragraph proposed.

Other than the above, the paper can be published as is.